# Bacterial Skin Microbiota of Seabass from Aegean Fish Farms and Antibiotic Susceptibility of Psychrotrophic *Pseudomonas*

**DOI:** 10.3390/foods12101956

**Published:** 2023-05-11

**Authors:** Ali Aydin, Mert Sudagidan, Zhanylbubu Mamatova, Mediha Nur Zafer Yurt, Veli Cengiz Ozalp, Jacob Zornu, Saraya Tavornpanich, Edgar Brun

**Affiliations:** 1Department of Food Hygiene and Technology, Faculty of Veterinary Medicine, Istanbul University-Cerrahpasa, Avcilar, Istanbul 34320, Turkey; 2KIT-ARGEM R&D Center, Konya Food and Agriculture University, Meram, Konya 42080, Turkey; 3Department of Medical Biology, Medical School, Atilim University, Golbasi, Ankara 06830, Turkey; 4Norwegian Veterinary Institute, 1433 Ås, Norway

**Keywords:** seabass, microbiota, fish farms, *Pseudomonas*, antibiotic resistance

## Abstract

Farming seabass (*Dicentrarchus labrax*) is an essential activity in the Mediterranean basin including the Aegean Sea. The main seabass producer is Turkey accounting for 155,151 tons of production in 2021. In this study, skin swabs of seabass farmed in the Aegean Sea were analysed with regard to the isolation and identification of *Pseudomonas*. Bacterial microbiota of skin samples (*n* = 96) from 12 fish farms were investigated using next-generation sequencing (NGS) and metabarcoding analysis. The results demonstrated that Proteobacteria was the dominant bacterial phylum in all samples. At the species level, *Pseudomonas lundensis* was identified in all samples. *Pseudomonas, Shewanella*, and *Flavobacterium* were identified using conventional methods and a total of 46 viable (48% of all NGS+) *Pseudomonas* were isolated in seabass swab samples. Additionally, antibiotic susceptibility was determined according to standards of the European Committee on Antimicrobial Susceptibility Testing (EUCAST) and Clinical and Laboratory Standards Institute (CLSI) in psychrotrophic *Pseudomonas*. *Pseudomonas* strains were tested for susceptibility to 11 antibiotics (piperacillin-tazobactam, gentamicin, tobramycin, amikacin, doripenem, meropenem, imipenem, levofloxacin, ciprofloxacin, norfloxacin, and tetracycline) from five different groups of antibiotics (penicillins, aminoglycosides, carbapenems, fluoroquinolones, and tetracyclines). The antibiotics chosen were not specifically linked to usage by the aquaculture industry. According to the EUCAST and CLSI, three and two *Pseudomonas* strains were found to be resistant to doripenem and imipenem (E-test), respectively. All strains were susceptible to piperacillin-tazobactam, amikacin, levofloxacin, and tetracycline. Our data provide insight into different bacteria that are prevalent in the skin microbiota of seabass sampled from the Aegean Sea in Turkey, and into the antibiotic resistance of psychrotrophic *Pseudomonas* spp.

## 1. Introduction

Seafood, especially fish, is an increasingly important component of human diets. Thus, aquaculture is an important source of food suitable for human consumption [1], and could provide a sustainable supply of affordable seafood to an increasing global population. Mediterranean marine aquaculture grew exponentially during the last decades of the 20th century, though at a slower pace over the past 20 years or so [2]. European seabass (*Dicentrarchus labrax*) is the 31st most-reared fish in worldwide aquaculture [3]. Seabass production increased by 2.9% in 2020 and reached 243,900 tons globally [4]. More than 95% of the world’s seabass and sea bream (*Sparus aurata*) production comes from aquaculture, of which, 97% accounts for the production in Mediterranean countries. Turkey and Greece are the primary producers, while Spain, France, Italy, Greece, and Turkey are the primary consumers [5].

Skin microbiota of fish species such as seabass have, however, hardly been investigated. To fill this knowledge gap, sampled seabass could be analyzed e.g., using next generation sequencing (NGS) whole genome sequencing and metabarcoding analysis. Such an approach would generate essential information on the profiles of both culturable and non-culturable microbial communities [6]. Furthermore, determining dominant microorganisms by NGS could contribute to the identification of pathogenic and/or potentially pathogenic bacteria in the aquaculture industry.

Although *Pseudomonas* species (including *P. aeruginosa*, *P. fluorescens*, *P. baetica*, *P. putida*, and *P. lundensis*) have been described as opportunistic human pathogens, many *Pseudomonas* species have also been associated with several diseases in farmed fish [7,8]. Additionally, psychrophilic *Pseudomonas* spp. cause spoilage of fishery products.

Apart from considerably limiting the success of aquaculture, the prevalence of fish diseases of microbial origin also necessitates the use of antibiotic treatments. Such treatments, particularly when applied without prudent justification, are known to cause the emergence of antibiotic-resistant bacteria [9]. Consequently, there is a continuous risk of the emergence of antibiotic resistance (AR) or multidrug resistance (MDR), i.e., the ability of a microorganism to withstand the action of one or more antimicrobial compounds [10]. Research has demonstrated the predominance and persistence of *Pseudomonas* spp. in, and on the surface of, seafood and in food processing plants, which reflects the ability of these microorganisms to withstand adverse conditions, including several antimicrobial treatments [11]. In addition, antibiotics are frequently used in the treatment of diseases in fish farming. Microbial communities on fish skin are highly variable, may be responsible for causing fish diseases, and may threaten the health of consumers [12]. Commonly, standard s of the European Committee on Antimicrobial Susceptibility Testing (EUCAST) [13] and Clinical and Laboratory Standards Institute (CLSI) [14] are used to determine the antibiotic susceptibility of bacteria in food intended for human consumption.

This study aimed to use NGS and metabarcoding analysis to determine the bacterial microbiota of seabass skin samples collected from fish farms in different parts of the Aegean Sea of Turkey. In addition, agar diffusion assays were performed to evaluate the antibiotic susceptibility against 11 antibiotics (piperacillin-tazobactam, gentamicin, tobramycin, amikacin, doripenem, meropenem, imipenem, levofloxacin, ciprofloxacin, norfloxacin, and tetracycline) from five antibiotics groups (penicillins, aminoglycosides, carbapenems, fluoroquinolones, and tetracyclines). Based on results from agar-disc diffusion assays and the E-test, Minimum Inhibitory Concentration (MIC) values were utilized to evaluate resistant psychrotrophic *Pseudomonas* strains in accordance with EUCAST and CLSI criteria [13,14].

## 2. Materials and Methods

### 2.1. Sampling

During June 2022, 96 seabass with an average weight of 300 g and average length of 220 mm were obtained from fish farms in 12 locations (8 samples per farm) in the Aegean Sea. These fish farms belonged to five different aquaculture companies and were labeled using capital letters with a numerical subscript (i.e., A1, A2, A3, B1, B2, C1, C2, D1, D2, E1, E2, and E3) (Figure 1). The collected fish were stored in styrofoam boxes containing aseptic ice and transported within 4–6 h in refrigerated vehicles (+4 °C) to the international market chain in Istanbul. The styrofoam boxes were opened immediately on arrival under aseptic conditions. The central temperature in the boxes was ≤+4 °C measured with a thermometer (Testo, Lenzkirch, Germany). Under the same conditions, the samples were taken by rubbing off the skin of the seabass with sterile swabs containing a transport liquid medium (Becton Dickinson, NJ, USA). The swabs were transported under refrigeration temperatures in thermal boxes (≤+4 °C) to the laboratory (Department of Food Science and Technology, Istanbul University-Cerrahpasa) for immediate analyses.

### 2.2. Next Generation Sequencing (NGS) and Metabarcoding Analysis

#### 2.2.1. Total DNA Extraction

Total DNA extraction was carried out directly from the swab samples by applying the phenol/chloroform/isoamyl alcohol method [15]. For this purpose, 2 mL swab samples were centrifuged at 14,000 rpm for 5 min at room temperature. The pellet was resuspended in 500 μL 1×TE buffer (10 mM Tris-HCl, 1 mM EDTA, pH 8.0) containing 5 mg/mL lysozyme (Applichem, Darmstadt, Germany) and the phenol/chloroform/isoamyl alcohol was applied. Finally, the extracted DNA samples were resuspended in 30 μL sterile deionized water and stored at −20 °C for amplicon PCR experiments in NGS studies.

#### 2.2.2. Next-Generation Sequencing

16S rRNA amplicon sequencing and DNA library preparation were carried out according to the 16S metabarcoding sequencing library preparation guide [16]. The primers for the amplicon PCR were F-primer: 5′-TCGTCGGCAGCGTCAGATGTGTATAAGAGACAGCCTACGGGNGGCWGCAG-3′ and R-primer: 5′-GTCTCGTGGGCTCGGAGATGTGTATAAGAGACAGGACTACHVGGGTATCTAACC-3′. Bacterial 16S rRNA V3-V4 gene regions were amplified using a KAPA HiFi HS kit (Roche, Mannheim, Germany). PCR products from each seabass sample were indexed with dual indexes using a Nextera^®^ XT Index Kit v2 Set-A (Illumina, San Diego, USA). All the amplicon PCR products and indexed amplicons were purified using AMPure XP magnetic beads (Beckman Coulter, Indianapolis, USA). The prepared equimolar proportions (10 nM) of the samples were pooled, and diluted to a 35 pM library containing 5% (*v*/*v*) PhiX control DNA (Illumina). Subsequently, a 20 μL library was loaded into an iSeq100 v1 cartridge. The sequencing was carried out using the iSeq100 system (Illumina) pair end read type and two reads of 151 bp read length.

#### 2.2.3. Metabarcoding Analysis

The sequencing reads from the 16S rRNA gene were analyzed using Silva NGS software version 138.1, VSEARCH 2.17.0, SINA v1.2.10 for ARB SVN (revision 21008), and BLASTn version 2.11.0+. Trimming of adapter sequences from short NGS read data was performed using Genious Prime software. The amplicons were clustered based on the sequence identity operational taxonomic unit (OTU) approach. Clustering Ward’s analysis was applied using the PAleontological STatistics (PAST) Software version 4.11 package (2022) at the genus levels in the seabass samples [17].

### 2.3. Isolation and Identification of Pseudomonas

*Pseudomonas* spp. isolation and identification were performed using the modified conventional TS EN ISO 13720 standard [18]. First, 250 µL of the swab sample containing each liquid medium was taken and placed in 2 mL of Pseudomonas Broth (Z699101 Merck, Darmstadt, Germany) and incubated at 22 ± 2 °C for 44 ± 4 h (Pre-enrichment). Subsequently, 0.1 mL of the suspension in Pseudomonas Broth was taken and spread onto Pseudomonas Agar (CM 559 Oxoid, Basingstoke, UK) containing Pseudomonas CFC Selective Supplement (SR103 Oxoid). The plates were incubated at 22 ± 2 °C for 44 ± 4 h. After incubation, suspected *Pseudomonas* spp. were transferred to Tryptic Soy Agar (CM 131, Oxoid, Basingstoke, UK) for purification. Biochemical tests such as Gram staining, oxidase test, catalase test, and fluorescence properties with UV light (365 nm) were applied to confirm *Pseudomonas* strains [9,18].

### 2.4. Determination of Antibiotic Susceptibility in Psychrotrophic Pseudomonas Strains

*Pseudomonas* strains were tested for antibiotic susceptibility using the agar disk diffusion method on Mueller–Hinton agar (CM 337 Oxoid) [19]. The plates were incubated at 22 ± 2 °C for 24 h. Eleven (11) different antibiotics were used: Piperacillin-tazobactam (Oxoid-CT1616, 30–6 µg), gentamicin (Oxoid-CT0024, 10 µg), tobramycin (Oxoid-CT0056, 10 µg), amikacin (Oxoid-CT0107, 30 µg), doripenem (Oxoid-CT1880, 10 µg), meropenem (Oxoid-CT0774, 10 µg), imipenem (Oxoid-CT0455, 10 µg), levofloxacin (Oxoid-CT1587, 5 µg), ciprofloxacin (Oxoid-CT0425, 5 µg), norfloxacin (Oxoid-CT0434, 10 µg) and tetracycline (Oxoid-CT0054, 30 µg) according to the CLSI [14] from five preferred antibiotic groups (penicillins, aminoglycosides, carbapenems, fluoroquinolones, and tetracyclines).

The E-test (Bioanalyse, Turkey) was applied to determine the Minimum Inhibitory Concentration (MIC) of *Pseudomonas* strains that were found to be resistant to antibiotics in the disc diffusion test. Results were evaluated according to the EUCAST [13] and CLSI [14] breakpoint tables.

## 3. Results and Discussion

### 3.1. NGS and Metabarcoding Analysis Results

Modern high-throughput methods have substituted conventional culture-based microbiological techniques, increasing our understanding of fish microbial communities throughout the production chain, from harvesting through storage distribution, until the end of shelf life [20]. In this study, the alpha diversity of bacteria was estimated to determine the diversity within samples, and the Shannon species diversity index values were determined using Silva NGS software (Table 1). This diversity index is a quantitative measure for estimating the number of different species in a given environment and their relative abundance [21]. This can be relevant for identifying the bacterial diversity in skin seabass samples because skin mucus harbors a complex bacterial community [22].

Metabarcoding analysis of 189,207 sequences from 96 seabass skin samples led to 123,391 OTUs, 39,737 clustered sequences, and 164,870 classified sequences. The results indicated that the phylum Proteobacteria was dominant in all seabass skin samples. The skin microbiota samples also contained bacteria belonging to the phyla Firmicutes and Bacteroidota (Figure 2). At the genus level, *Pseudomonas* was the dominant genus among the 96 seabass swab samples. (Figure 3). Additionally, *Shewanella*, *Acinetobacter*, and *Flavobacterium* were also among the most prevalent genera (Figure 3). Similar results were reported from the Bodrum coast in seawater, Mugla [23]. The genus *Pseudomonas* is considered to be an important fish pathogen as it comprises some (sub) species which are opportunistic pathogens to humans [23]. Another study dedicated to examining the microbiota of whole and filleted seabass [20] presented results similar to those we obtained. *Pseudomonas* was dominant in seabass samples, based on the 16S rRNA metabarcoding analysis, followed by the presence of *Shewanella*. Among animal food products, fish are the most vulnerable to bacterial spoilage and *Shewanella* has previously been reported as a main contributor in the microbiota of spoiled seafood, such as hake fillets [24]. Additionally, *Shewanella* was the dominant genus in MAP-stored seabass fillets, but its relative abundance declined dramatically towards the end of the products’ shelf life [19]. *Acinetobacter* are abundant in aquatic environments and frequently isolated from the skin and gills of fresh fish [25]. In a previous study, *Acinetobacter* were the dominant bacteria in seabass fillets [20] and rainbow trout samples [26]. However, *Acinetobacter* are not recognized as important spoilage bacteria [27] as they cannot hydrolyze fish proteins and are thus, a weak producer of biogenic amines, as well as a weak degrader of ATP-related compounds [28].

*P. lundensis* was identified by NGS analysis of all seabass samples. Similar to our results, Elbehiry et al. [29] reported that, in red meat samples, *P. lundensis* was the dominant species. Pseudomonads are highly opportunistic and may become a highly threatening fish pathogen causing serious illness including ulcerative syndrome and hemorrhagic septicemia [30]. *Enterococcus* were found in 15 seabass samples, of which samples S1, S4, and S5 were sampled from the same fish farm. The other *Enterococcus*-containing samples were S11, S34, S45, S53, S54, S57, S64, S65, S66, S67, S68, and S69 identified from four different fish farms (D2 and C2 located in Mugla, A2 and E2 located in Izmir). Detection of *Enterococcus* spp. in sea bass skin samples may indicate fecal contamination in seawater.

The highest Shannon diversity index in this study (7.25) was obtained for samples S3 and S13, indicating that these samples had the highest diversity of skin microbiota. The S28 sample contained the lowest species diversity with a value of 6.16 (Table 1). Ward’s analysis demonstrated that two main clusters were present at the genus level (Figure 4). The composition of the microbiota, however, did not cluster at the genus level. This might be attributable to differences in the composition of the fish skin microbiomes between individual fish from the same population and differences between the skin microbiome and the surrounding water [6].

Foodborne pathogens such as *Salmonella*, *Escherichia,* and *Mycobacterium* genera were not found in the samples. On the other hand, *Vibrio* (*V*.) *ordalii* was detected in three seabass swab samples (numbers 65, 66, and 67) originating from E-2 fish farms in Izmir. Similarly, many researchers have reported *V. ordalii* from seabass in the Aegean Sea [31,32], including Izmir [33]. Bacterial infections most frequently detected in cultured seabass and gilthead sea bream are caused by bacteria belonging to the family *Vibrionaceae*. Associated losses have been reported with *Vibrionaceae* in many fish species, including seabass, sea bream, and salmonid species etc. [34].

### 3.2. Temperature Measurement of Seabass Samples in Styrofoam Boxes Containing Ice

The lowest average temperature was 1.7 °C in the samples from the fish farm B2 located in Izmir, and the highest temperature was 3.4 °C in the samples from the fish farms A3 (Izmir) and E2 (Izmir) (Table 2). The average and standard deviation of the inner temperature of seabass samples were 2.58 ± 0.53 °C. In addition, the internal temperature values measured in all fish samples were below +4 °C. Similarly, a study reported the internal temperature of iced styrofoam-packaged seabass from the Aegean Sea to be 4.15 ± 1.12 °C [35]. The extension of shelf life by chilling is essentially due to the reduction in the growth rate and metabolic activity of spoilage microorganisms such as *Pseudomonas* spp. [35] and *Acinetobacter* spp. *Acinetobacter* species have been found in great abundance in fresh seabass at 12 °C [19] and fish fillets at 10 °C [36], and were the dominant species at the end of the shelf life of rainbow trout stored aerobically at 4 °C [25]. Indeed, upon storage the psychrophilic bacteria proliferated slowly and dominated the mesophilic load, as the low temperature favored their growth [37]. Similar to our study, Syropoulou et al. [38] reported that *Pseudomonas* spp. were found from the beginning of shelf life, whilst in seabass products from Greece, *Shewanella* were detected at later storage stages.

### 3.3. Isolation of Psychrotrophic Pseudomonas spp. in Seabass Swab Samples using Conventional Methods

In total, 46 seabass swab samples (48%) were positive for psychrotrophic *Pseudomonas* strains isolated with the conventional ISO method [18] (Table 2). *Pseudomonas* strains were isolated from four fish farms in Izmir, i.e., A2 (*n* = 6), E2 (*n* = 6), E3 (*n* = 6), and A3 (*n* = 5), and farm C1 (*n* = 5) in Mugla. The cultivation-based method will detect live *Pseudomonas* strains, which is an important characteristic when compared to NGS and metabarcoding methods that are used in the detection of DNA fragments and DNA structures, as these do not necessarily indicate the presence of living bacteria [39].

### 3.4. Antibiotic Susceptibility of Pseudomonas spp. Using Disc Diffusion

Susceptibility to 11 antibiotics was tested among 46 viable *Pseudomonas* spp. isolates. Some of the strains (13/46; 28.3%) were found to be resistant to doripenem, according to EUCAST [13] and CLSI [14] (Table 3).

Thirty (65.2%) *Pseudomonas* strains were susceptible to all antibiotics according to the CLSI [14]. On the other hand, thirty-three (71.7%) *Pseudomonas* strains were susceptible to all antibiotics according to the EUCAST [13]. All *Pseudomonas* strains from A1 (Izmir), B1 (Izmir), and E1 (Izmir) fish farms were susceptible to all antibiotics (Table 4).

Sixteen (34.8%) *Pseudomonas* strains were resistant to more than one antibiotic based on the CLSI [14]. Eight (17.4%) *Pseudomonas* strains were resistant to one antibiotic only, including carbapenem (doripenem) and aminoglycoside group (tobramycin). Six *Pseudomonas* strains were resistant to doripenem, and two strains were resistant to tobramycin based on the CLSI [14]. However, only five (10.9%) *Pseudomonas* strains were resistant to two antibiotics, according to the CLSI [14]. All *Pseudomonas* strains from fish farms in Izmir [(A2; *n* = 1) and (A3; *n* = 1)] and Mugla [(C1; *n* = 1) and (D1; *n* = 1)] were resistant to doripenem and imipenem (carbapenem group). In addition, one strain originating from D2 fish farms (Mugla) was found to be MDR to doripenem, imipenem, and meropenem, all included in the carbapenem group, based on the CLSI (Table 4).

Thirteen (28.3%) *Pseudomonas* strains were found to be resistant to several antibiotics according to EUCAST [13], seven (13.4%) to only one antibiotic, including carbapenem (doripenem) and fluoroquinolones (norfloxacin) group. Four *Pseudomonas* strains were resistant to doripenem, and two strains to tobramycin according to the EUCAST standard [13]. Additionally, only two (4.3%) *Pseudomonas* strains isolated from Izmir (A3 and B2) were resistant to two antibiotics, according to the EUCAST standard [13]. Moreover, five *Pseudomonas* strains originating from Izmir (A2), and Mugla (C1, C2, D1, and D2) fish farms were found to be MDR to doripenem, imipenem, and meropenem including carbapenem group based on the EUCAST [13] (Table 4).

*Pseudomonas* spp. have been identified as primarily invasive or opportunistic pathogens for many organisms and this genus has also grown in importance in terms of antimicrobial resistance [9]. Many researchers have evaluated the antimicrobial sensitivity of *Pseudomonas* species isolated from fish, and have reported them as MDR, based on their resistance to ampicillin, cefotaxime, aztreonam, trimethoprim-sulfamethoxazole, nitrofurantoin and other groups of antimicrobials [9,40]. Recently, Rezgui et al. [41] showed an abundance of antibiotic-resistant bacteria isolated from the gills and intestinal tract of seabass and sea bream. The antibiotic-resistant bacteria belong to several species of the genera *Pseudomonas*, *Vibrio*, *Aeromonas*, and *Enterobacterales*. They were resistant to tetracycline and penicillin, which are commonly used in treating infections in animals and humans. In another study, almost all *Pseudomonas* strains were resistant to penicillins (ampicillin), macrolides (erythromycin, clindamycin), sulfonamides (trimethoprim-sulphamethoxazole-), and chloramphenicol [9]. We report similar results, i.e., that the *Pseudomonas* strains were susceptible to penicillins (piperacillin-tazobactam), aminoglycosides (amikacin and gentamycin), fluoroquinolones (levofloxacin, norfloxacin), and tetracyclines (tetracycline, ciprofloxacin) based on the CLSI [14]. Likewise, a study reported that enrofloxacin, oxytetracycline, and ciprofloxacin were found to be effective antibiotics against fish disease agents such as *Pseudomonas* spp., *Vibrio* spp. and *Staphylococcus* spp. in Turkey [42]. On the other hand, all *P. fluorescence* strains isolated from fish were resistant to piperacillin, ceftazidime, and cefepime in Egypt [43]. In the present study, psychrotrophic *Pseudomonas* strains were partially resistant (based on the EUCAST and CLSI) to antibiotics commonly used in fish farms. This fact should be carefully addressed in the context of the environmental spread of antibiotic resistance.

According to the CLSI, psychrotrophic *Pseudomonas* strains showed different resistance patterns to doripenem (28.3%), imipenem (13%), tobramycin (4.3%), and meropenem (2.3%). Similarly, *Pseudomonas* were resistant to doripenem (28.3%), imipenem (13%), meropenem (10.9%) and ciprofloxacin (2.2%) based on the EUCAST. In total, *Pseudomonas* strains resistant to nine antibiotics were isolated from nine different fish farms [A2 (*n* = 1), A3 (*n* = 2), B2 (*n* = 2), C1 (*n* = 1), C2 (*n* = 1), D1 (*n* = 1), D2 (*n* = 1), E2 (*n* = 1), and E3 (*n* = 2)]. *Pseudomonas* strains were resistant to the same antibiotics (imipenem, meropenem, and doripenem) (Table 4). Additionally, one *Pseudomonas* strain belonging to B2 (*n* = 1) fish farm showed resistance to ciprofloxacin and doripenem based on the EUCAST [13]. Finally, five *Pseudomonas* strains resistant to three antibiotics were identified according to the EUCAST [13]. These strains originated from five different fish farms: A2 (*n* = 1, Izmir), C1(*n* = 1, Mugla), C2 (*n* = 1, Mugla), D1 (*n* = 1, Mugla), and D2 (*n* = 1, Mugla). Fish diseases are limiting factors in fish production, causing high mortality, especially in hatcheries, which affects profit negatively [29]. Antibacterial therapy is often chosen as the way to control bacterial disease outbreaks that pose economic challenges [43]. Additionally, antibiotic resistance is one of the most significant challenges to human health and food security [28]. Some studies are available on antibiotic susceptibility in human pathogenic bacteria, including *Pseudomonas* spp. [44].

### 3.5. MICs of Psychrotrophic Pseudomonas spp.

*Pseudomonas* strains that had shown resistance to antibiotics in the disc diffusion assay were selected for examination using the E-Test (gradient diffusion method) to determine their MIC (Table 5). From the 13 strains that showed resistance to doripenem in the disc diffusion test, two had an MIC exceeding the threshold ≥8 g/mL for antibiotic resistance (12 and 125 g/mL; the latter isolate originated from farm A3 in İzmir). For imipenem, three out of six isolates resistant according to disc-diffusion assay were confirmed as resistant by E-test. The MIC of these three resistant strains was >32 μg/mL. All these isolates originated from farms A3 (in Izmir), C1, and D1 (both in Mugla). Similarly, isolates resistant to tobramycin, meropenem, or ciprofloxacin according to the disc diffusion assay, were classified as susceptible based on the E-test MIC [13,14]. Only one *Pseudomonas* strain from C1 fish farms (Sample no. 24) was resistant to doripenem and imipenem, as assessed by MIC determination.

The different results obtained by the gradient diffusion (E-test) and the disc diffusion methods for *Enterobacterales* and *Pseudomonas aeruginosa* strains are not unexpected since the E-test generally performs better [45]. Despite the different outcomes from different methods, our results are in line with reports on antimicrobial resistance in *Pseudomonas* and *Escherichia coli* in general. The European Antimicrobial Resistance Surveillance Network reported on samples from human patients in 2017, of which, 30.8% of the *Pseudomonas aeruginosa* strains isolated were resistant to at least one of the antimicrobial groups under regular surveillance (fluoroquinolones, aminoglycosides, and carbapenems) [46]. Moreover, the European Centre for Disease Prevention and Control has shown significant increments in the percentage of antibiotic-resistance among pathogenic bacteria, such as carbapenem-resistance in *Pseudomonas aeruginosa* and *Acinetobacter* spp. in several countries in the European region of concern [47].

With respect to fish, a study from Egypt reported that *Pseudomonas aeruginosa* and *E. coli* strains were resistant to third-generation cephalosporin and last-resort carbapenems isolated from Nile tilapia [41]. Interestingly, 29.7% of *P. fluorescens* strains isolated showed MDR, especially to penicillin and cephalosporin groups [41].

## 4. Conclusions

Results from this study show that psychrotrophic *Pseudomonas* were the dominant bacterial species in seabass skin samples from 12 selected fish farms in the Aegean Sea. Ninety-six fish were sampled by skin swab, and in all samples, NGS analysis indicated the presence of *Pseudomonas*. Viable isolates were cultured from 46 of these samples. Testing the isolates against 11 different antibiotics (five main groups), showed that all samples were susceptible to piperacillin-tazobactam, gentamicin, amikacin, levofloxacin, norfloxacin, and tetracycline. Based on the CLSI, the isolates from across the farms showed various resistance patterns to the carbapenem group [doripenem (28.3%), imipenem (13%), and meropenem (2.3%)] and aminoglycosides [tobramycin (4.3%)]. Using the EUCAST standard, there was additional resistance to doripenem (28.3%), imipenem (13%), meropenem (2.3%), and ciprofloxacin (2.2%). MDR was found among three *Pseudomonas* strains from Mugla (D = 2) based on the CLSI and five *Pseudomonas* strains based on the EUCAST criteria (disc diffusion method). Three farms with six isolates showed no antibiotic resistance based on EUCAST and CLSI criteria.

This study has shown that resistance to a broad range of antibiotics prevails in *Pseudomonas* from the selected farms. As the farms were chosen without looking at their histories of disease and antibiotic use, our results may indicate a representative situation for the industry in the region. This should, however, be confirmed in a broader study, including records of antibiotic use at the farm level.

The use of antibiotics is generally regarded as the main driver for developing resistance. Exposure to antibiotics may be due to own use or external exposure. The industry uses antibiotics for prophylactic and therapeutic treatments to keep farmed fish free of diseases. Prudent use of antibiotics is therefore essential also for the aquaculture industry to minimize antibiotic resistance and the spread of resistant bacteria or genes to the environment. Ultimately, this will serve consumer protection and lead to a more efficient application of antibiotics in human therapy.

## Figures and Tables

**Figure 1 foods-12-01956-f001:**
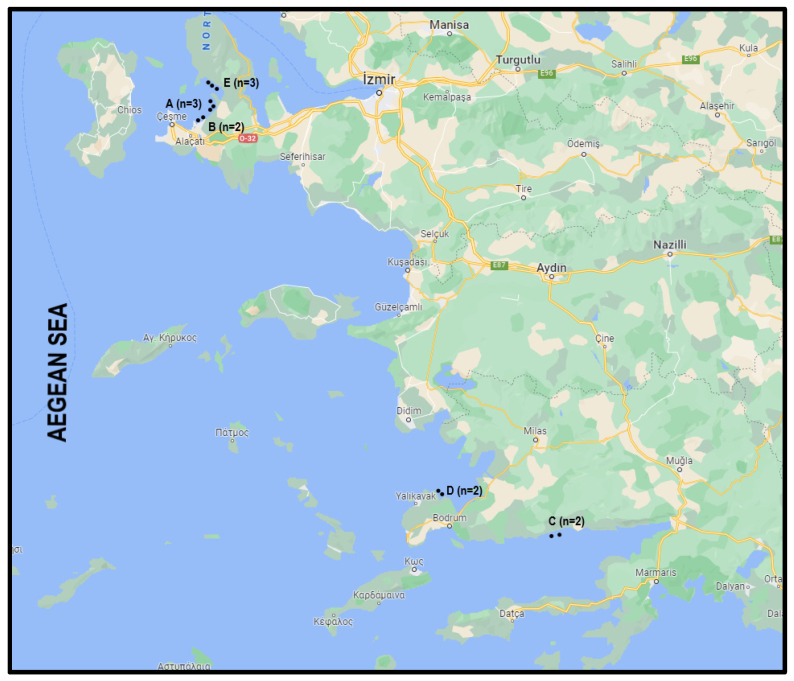
Seabass aquaculture companies and fish farms locations in the Aegean Sea. (Aquaculture Company A: three fish farms in Izmir; Aquaculture Company B: two fish farms in Izmir; Aquaculture Company C: two fish farms in Mugla; Aquaculture Company D: two fish farms in Mugla; and Aquaculture Company E: three fish farms in Izmir).

**Figure 2 foods-12-01956-f002:**
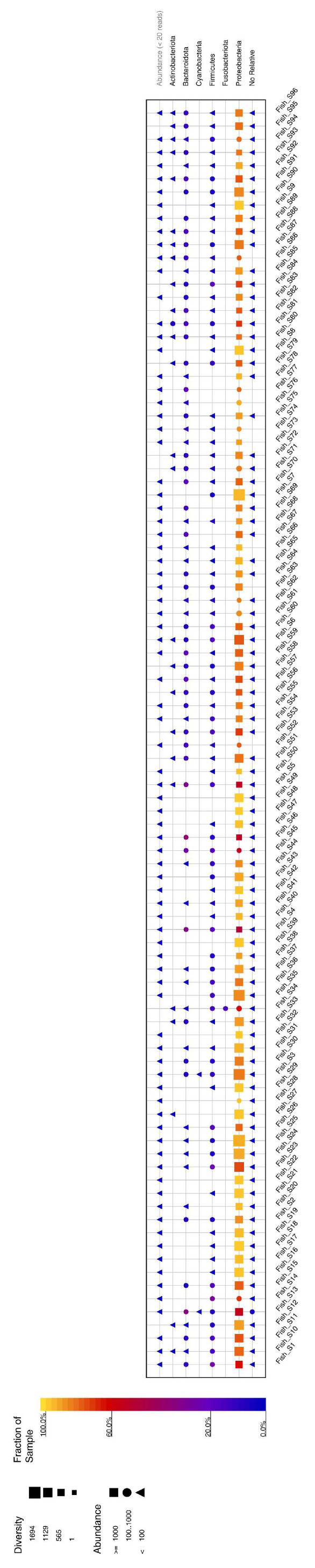
Distribution of bacterial communities in seabass swab samples at the phylum level.

**Figure 3 foods-12-01956-f003:**
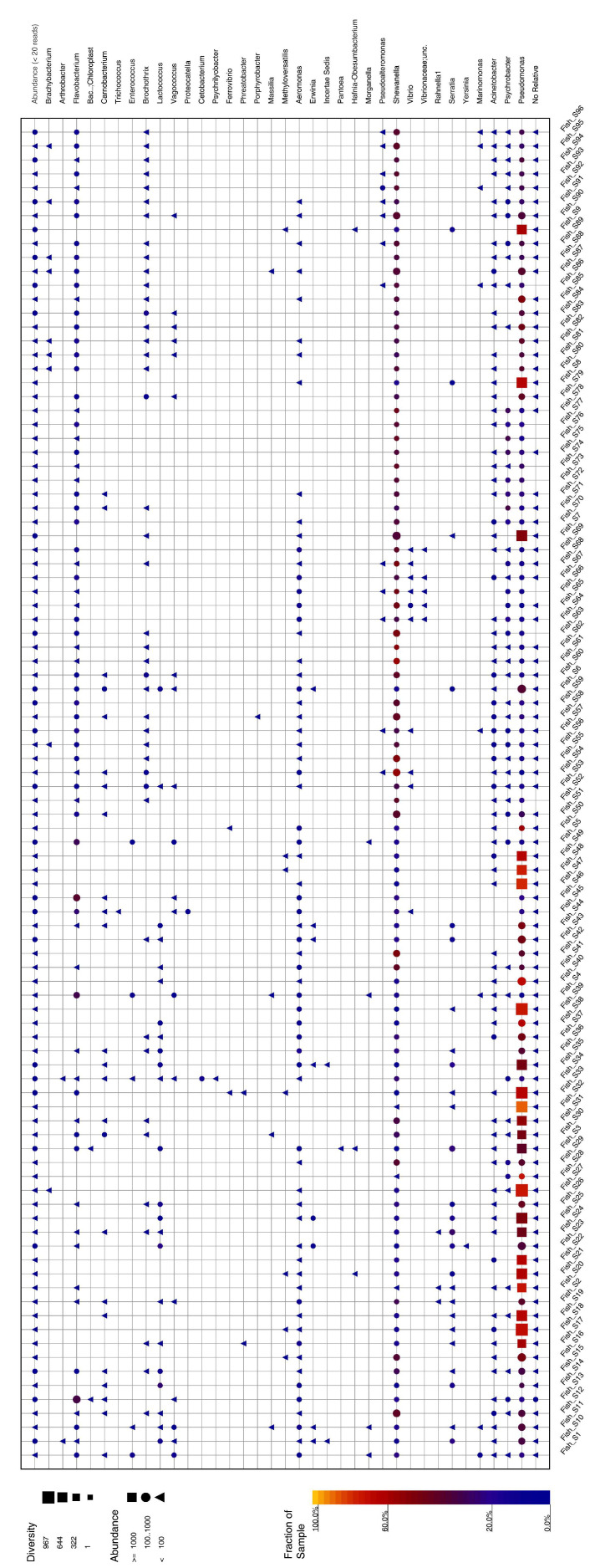
Distribution of bacterial communities in seabass swab samples at the genus level.

**Figure 4 foods-12-01956-f004:**
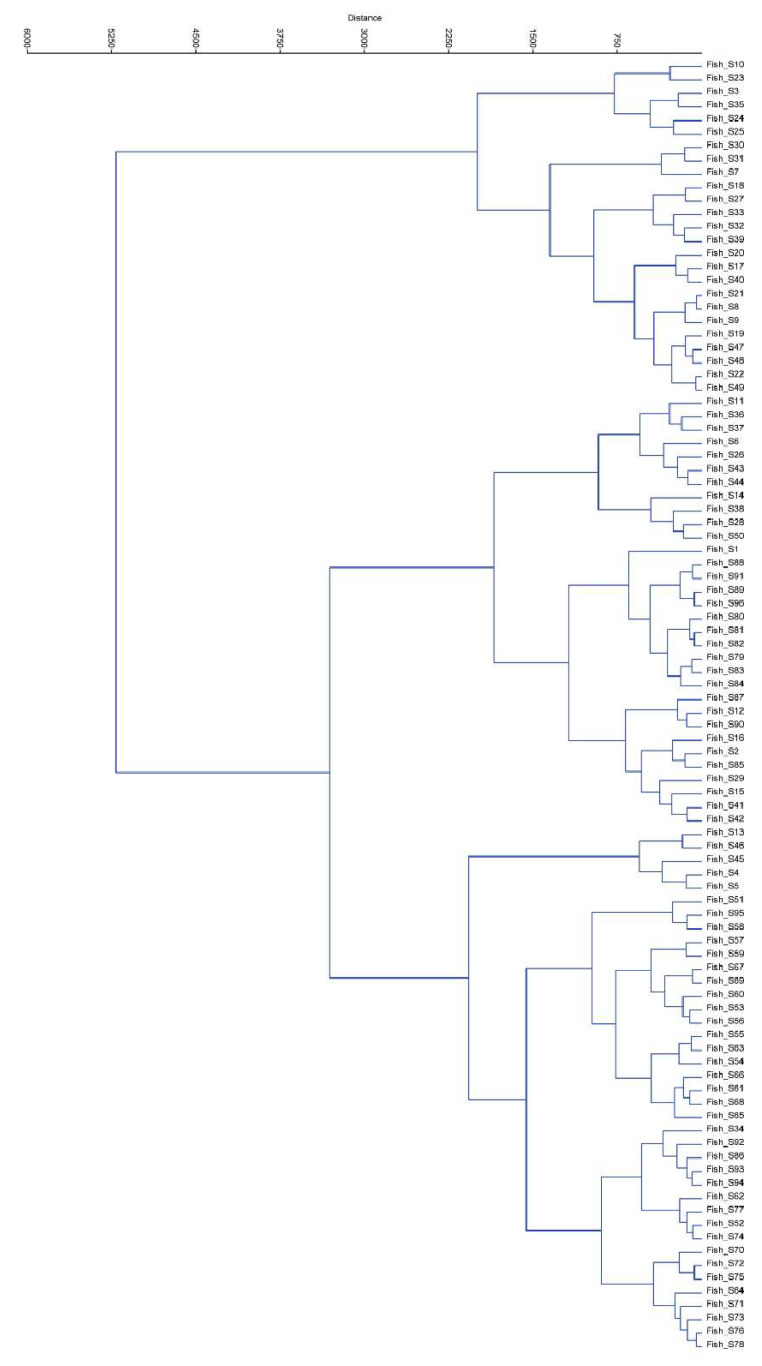
Dendrogram based on Ward’s method of clustering.

**Table 1 foods-12-01956-t001:** Shannon species diversity index values * of seabass skin samples.

Company Code	Sample Name	Shannon Index	Company Code	Sample Name	Shannon Index	Company Code	Sample Name	Shannon Index
A1	Fish_S1	7.02	E1	Fish_S33	7.03	E2	Fish_S65	6.73
	Fish_S2	6.84		Fish_S34	6.83		Fish_S66	6.51
	Fish_S3	7.25		Fish_S35	7.14		Fish_S67	6.88
	Fish_S4	7.11		Fish_S36	6.93		Fish_S68	6.61
	Fish_S5	7.10		Fish_S37	6.95		Fish_S69	6.78
	Fish_S6	7.21		Fish_S38	6.58		Fish_S70	6.94
	Fish_S7	6.99		Fish_S39	6.82		Fish_S71	6.66
	Fish_S8	6.88		Fish_S40	6.58		Fish_S72	6.82
B1	Fish_S9	6.89	D2	Fish_S41	6.76	B2	Fish_S73	6.56
	Fish_S10	7.14		Fish_S42	6.68		Fish_S74	6.29
	Fish_S11	7.14		Fish_S43	6.88		Fish_S75	6.62
	Fish_S12	7.00		Fish_S44	6.84		Fish_S76	6.44
	Fish_S13	7.25		Fish_S45	6.92		Fish_S77	6.38
	Fish_S14	6.66		Fish_S46	7.02		Fish_S78	6.52
	Fish_S15	7.12		Fish_S47	6.65		Fish_S79	6.78
	Fish_S16	6.91		Fish_S48	6.55		Fish_S80	6.52
C1	Fish_S17	6.86	C2	Fish_S49	6.82	E3	Fish_S81	6.83
	Fish_S18	6.88		Fish_S50	6.33		Fish_S82	6.73
	Fish_S19	6.87		Fish_S51	7.06		Fish_S83	6.80
	Fish_S20	6.53		Fish_S52	6.30		Fish_S84	6.75
	Fish_S21	6.92		Fish_S53	7.01		Fish_S85	6.69
	Fish_S22	6.85		Fish_S54	6.79		Fish_S86	6.61
	Fish_S23	7.13		Fish_S55	6.70		Fish_S87	7.00
	Fish_S24	7.02		Fish_S56	6.84		Fish_S88	6.90
D1	Fish_S25	7.16	A2	Fish_S57	6.99	A3	Fish_S89	6.87
	Fish_S26	6.82		Fish_S58	7.00		Fish_S90	7.08
	Fish_S27	6.68		Fish_S59	6.98		Fish_S91	6.94
	Fish_S28	6.16		Fish_S60	6.81		Fish_S92	6.69
	Fish_S29	6.75		Fish_S61	6.65		Fish_S93	6.66
	Fish_S30	6.95		Fish_S62	6.34		Fish_S94	6.61
	Fish_S31	6.87		Fish_S63	6.87		Fish_S95	6.96
	Fish_S32	6.52		Fish_S64	6.74		Fish_S96	6.89

* The higher the index values, the more diverse the species in the habitat.

**Table 2 foods-12-01956-t002:** Temperature of seabass samples and verification of viable psychrotrophic *Pseudomonas* strains after Next Generation Sequencing analysis (NGS) using conventional methods [18].

Company Code and Fish Farm Number	Temperature Measurement of Seabass Samples (°C)	Samples with DNA Fragments from *Pseudomonas*	Number Samples with Viable *Pseudomonas* Strains (out of NGS Positive Samples)
1A1	2.3	8	3
A2	2.7	8	6
A3	3.4	8	4
2B1	2.7	8	1
B2	1.7	8	2
3C1	2.1	8	5
C2	2.3	8	4
4D1	3.2	8	4
D2	2.5	8	4
5E1	1.9	8	2
E2	3.4	8	6
E3	2.8	8	5
Totally	Total x-Sx 2.58 ± 0.53	96	46

^1^ Fish Company A: A1–A3, three different fish farms of fish company A in Izmir Province; ^2^ Fish Company B: B1–B2, two different fish farms of fish company B in Izmir Province; ^3^ Fish Company C: C1–C2, two different fish farms of fish company C in Mugla Province; ^4^ Fish Company D: D1–D2, two different fish farms of fish company D in Mugla Province; ^5^ Fish Company E: E1–E3, three different fish farms of fish company E in Izmir Province.

**Table 3 foods-12-01956-t003:** Clinical and Laboratory Standards Institute (CLSI) and the European Committee on Antimicrobial Susceptibility Testing (EUCAST) as assessed using the disc diffusion method of psychrotrophic *Pseudomonas* strains (*n* = 46) [Resistant (“R”); Intermediate susceptibility (“I”) or Susceptible (“S”)].

Antibiotic Groups	Name of Antibiotics	Distribution of *Pseudomonas* Strains according to CLSI			Distribution of *Pseudomonas* Strains According to EUCAST	
		R (%)	I (%)	S (%)	R (%)	S (%)
Penicillins	Piperacillin-tazobactam 30 µg	-	-	46 (100)	-	46 (100)
Aminoglycosides	Gentamicin 10 µg	-	-	46 (100)	*n* *	*n*
	Tobramycin 10 µg	2 (4.3)	-	44 (95.7)	*n*	*n*
	Amikacin 30 µg	-	-	46 (100)	-	46 (100)
Carbapenems	Doripenem 10 µg	13 (28.3)	-	33 (71.7)	13 (28.3)	33 (71.7)
	Meropenem 10 µg	1 (2.3)	4 (8.6)	41 (89.1)	5 (10.9)	41 (89.1)
	Imipenem 10 µg	6 (13)	2 (4.4)	38 (82.6)	6 (13)	40 (87)
Fluoroquinolones	Levofloxacin 5 µg	-	-	46 (100)	-	46 (100)
	Ciprofloxacin 5 µg	-	1 (2.2)	45 (97.8)	1 (2.2)	45 (97.8)
	Norfloxacin 10 µg	-	-	46 (100)	*n*	*n*
Tetracyclines	Tetracycline 30 µg	-	-	46 (100)	*n*	*n*

* *n*: A breakpoint value of this antibiotic is not available in the CLSI standard.

**Table 4 foods-12-01956-t004:** Distribution of susceptible, resistant, and multidrug resistant *Pseudomonas* spp. according to the Clinical and Laboratory Standards Institute (CLSI) [14] and the European Committee on Antimicrobial Susceptibility Testing (EUCAST) [13].

Company Name (Number of Isolated Strains per Farm)	CLSI				EUCAST			
	Number of Susceptible Strains	Number of *Pseudomonas* Strains Resistant to One Antibiotic	Number of *Pseudomonas* Strains Resistant to Two Antibiotics	* Number of MDR Strains	Number of Susceptible Strains	Number of *Pseudomonas* Strains Resistant to One Antibiotic	Number of *Pseudomonas* Strains Resistant to Two Antibiotics	Number of MDR Strains
A1 (*n* = 3)	3	-	-	-	3	-	-	-
A2 (*n* = 6)	5	-	1 (DOR **,IPM)	-	5	-	-	1 (DOR,IPM,MEM)
A3 (*n* = 4)	2	1 (DOR)	1 (DOR,IPM)	-	2	1 (DOR)	1 (DOR, IPM)	-
B1 (*n* = 1)	1	-	-	-	1	-	-	-
B2 (*n* = 2)	0	1 (DOR), 1 (TOB)	-	-	-	1 (TOB)	1 (CIP,DOR)	-
C1 (*n* = 5)	3	1 (TOB)	1 (DOR,IPM)	-	3	1 (TOB)	-	1 (DOR,IPM,MEM)
C2 (*n* = 4)	3	-	1 (DOR,IPM)	-	3	-	-	1 (DOR,IPM,MEM)
D1 (*n* = 4)	2	1 (DOR)	1 (DOR,IPM)	-	3	-	-	1 (DOR,IPM,MEM)
D2 (*n* = 4)	1	-	-	3 (DOR, IPM, MEM)	3	-	-	1 (DOR,IPM,MEM)
E1 (*n* = 2)	2	-	-	-	2	-	-	
E2 (*n* = 6)	5	1 (DOR)	-	-	5	1 (DOR)	-	
E3 (*n* = 5)	3	2 (DOR)	-	-	3	2 (DOR)	-	-

* Number of multidrug (3 or more) resistant *Pseudomonas* spp. ** Abbreviated Name of antibiotics (DOR: Doripenem; IPM: Imipenem; TOB: Tobramycin; MEM: Meropenem, and CIP: Ciprofloxacin).

**Table 5 foods-12-01956-t005:** The Minimum Inhibitory Concentrations (MIC), as assessed by E-Test, for four antimicrobial agents against *Pseudomonas* strains isolated from sea bass samples.

Group	Antimicrobial	Tested *n* = 22	MIC (µg/mL), *n* = 22										Resistant Isolates, *n* = 5
			0.012–0.025	0.026–0.50	0.051–0.999	1–1.5	3	4	6	12	>32	125	
Carbapenems	Doripenem ^1^	13		5	1			2	3	1		1	2
	Meropenem ^1^	1				1							0
	Imipenem ^1^	6	1	1				1			3		3
Aminoglycosides	Tobramycin ^2^	2					1		1				0
Fluoroquinolones	Ciprofloxacin ^3^	1	1										0

*n* = number of isolates; ^1^ = MIC ≥ 8 µg/mL indicates antimicrobial resistance according to CLSI and EUCAST; ^2^ = MIC ≥ 16 µg/mL indicates antimicrobial resistance according to CLSI; ^3^ = MIC ≥ 2 µg/mL indicates antimicrobial resistance according to EUCAST.

## Data Availability

The data presented in this study are available on request from the corresponding author.

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
