# Peer review of "Bacterial Skin Microbiota of Seabass from Aegean Fish Farms and Antibiotic Susceptibility of Psychrotrophic Pseudomonas"

_foods, 2023, doi:10.3390/foods12101956_

Round 1
Reviewer 1 Report
Dear Authors,
Please find the manuscript file that includes all comments. I hope you find the suggestions beneficial.

Author Response
We considered revising table 5 and section 3.5 again for clarity in manuscript.
Reviewer 1:
Page 1 Line 1: Title Use “Skin Tissue of” instead of “Surface” and use “collected” instead of “originated”
Answer (Revised paper, Page 0 Line 2): Corrected
Page 1 Line 15: The sentence is not understandable
Answer (Rev. paper, Page 0 Line 16): Corrected
Page 1 Line 16: The sentence is irrelevant
Answer (Rev. paper, Page 0 Line 16): This sentence was deleted.
Page 1 Line 17: There is no study or result on the pathogenic potential of isolated Pseudomonas species. Unless confirmation for pathogenic characteristic, the pathogenic potential should not be used
Answer (Rev. paper, Page 0 Line 18): According to the review recommendation, we removed “pathogenic potential” words in the paper.
Page 1 Line 18: This is an informative sentence. one of previous sentence is method and this sentence should be moved before previous one
Answer (Rev. paper, Page 0 Line 17): According to the review recommendation, this sentence was adjusted.
Page 1 Line 19: “bacterial microbiota of seabass skin samples,”
Answer (Rev. paper, Page 0 Line 19): Corrected
Page 2 Line 47-55: This section is irrelevant.
Answer (Rev. paper, Page 1 Line 47): Corrected (This section was totally rewritten).
Line 57: Before starting this section, authors need to make a logical flow on the importance of bacterial microbiota, determination of community and the importance of NGS Technology.
Answer (Rev. paper, Page 1 Line 55-58): Corrected (This paragraph and the previous paragraph were combined).
Page 2 Line 64: There is a requirement to connection between microbiota and Pseudomonas.
Answer (Rev. paper, Page 1 Line 57-63): Corrected
Page 2 Line 76: Pseudomonas should be italic in whole manuscript.
Answer (Rev. paper, Page 1 Line 70): Corrected. Manuscript is re-checked. All the name of bacteria italicized in manuscript.
Page 2 Line 82-86: This is a repetition of abstract.
Answer (Rev. paper, Page 1 Line 79-87): This sentence presented the aim of this study. Therefore we used this sentence in the abstract and introduction sections separately.
Page 2 Line 87: References should not be included here, the aim of this study.
Answer (Rev. paper, Page 1 Line 87): Corrected.
Page 3 Line 99: Using skin instead of surface should be more suitable. Surface can be used for materials. For animals and humans or other living organisms, authors should use skin instead of surface in the title, abstract, introduction and whole manuscript.
Answer (Rev. paper, Page 2 Line 99): We used “skin” instead of” surface” in whole manuscript.
Page 4 Line 137: This is a result and table 1 represent results of the study. It should be moved to the results section.
Answer (Rev. paper, Page 4 Line 186): According to the review recommendation, Table 1 moved the Results and Discussion section.
Page 4 Line 142: This is a result and it should be moved to the results section
Answer (Rev. paper, Page 5 Line 209): Figure 2 moved the Results and Discussion section. Figure 2 and figure 3 numbers have been swapped.
Page 5 Line 154: This is for result section.
Answer (Rev. paper, Page 4 Line 186): Table 1 moved the Results and Discussion section.
Page 6 Line 158: Figure 2 belongs for result, should be moved
Answer (Rev. paper, Page 6 Line 214): Figure 2 changed as Figure 3 and moved the Results and Discussion section.
Page 6 Line 159: Authors mentioned that antibiotic susceptibility was determined for psychrophilic Pseudomonas species. However, they used 35C in disc diffusion test. If they use only psychrophilic species, they need to use 22C for 48h or 28C for 24+-2h. In this regard, they need to delete psychrophilic mentions.
Answer (Rev. paper, Page 3 Lines 146): In the manuscript, it is written: The plates were incubated at 22±2°C for 24 h.
Page 7 Line 194: The important studies should be used for supporting spoile. Fish are one of the most vulnerable animals to bacterial spoilage. In this respect bacterial spoilage should be mentioned by authors. The some of similar articles presented:
Answer (Rev. paper, Page 4 Line 199-200): We added this sentences in this paper.
Page 8 Line 205: Please use “P. lundensis” after first usage.
Answer (Rev. paper, Page 7 Line 217): Corrected.
Page 8 Line 214: Could you please explain in detail why you used Shannon diversity index and what means of this index exactly?
Page 8 Line 225: Please use all bacteria names sort name after first full name usage (V. ordalii)
Answer (Rev. paper, Page 7 Line 237): Corrected.
Page 10 Line 231: Delete “Central temperature measurements recorded under aseptic conditions are presented 231 [Table 2]. In this context,”
Answer (Rev. paper, Page 8 Line 251): Deleted.
Page 10 Line 225: Table 2 and table 3 should be merged.
Answer (Rev. paper, Page 9 Line 273): Table 2 and table 3 combined as a Table 2.
Page 11 Line 272: Please use a correct citation. Additionally, CLSI (2021), the reference number is 14, does not direct a correct evaluation methodology. The reference number 14 is "Performance Standards for Antimicrobial Susceptibility Testing". This method is a standard for antimicrobial susceptibility testing not for evaluation.
Answer (Rev. paper, Page 3 Line 156): This standard included evaluation methodology. If the reviewer can access the relevant standard, he will have access to the evaluation methodology (Relevant standard can be shared) (Reference 19; CLSI, Clinical and Laboratory Standards Institute. M100-Ed31 Performance Standards for Antimicrobial Susceptibility Testing. 31st Edition, 2021. ISBN: 978-1-68440-105-5).
Page 11 Line 272 Table 4: As much as I see, I could not find any breakpoints on CLSI (2021), please check.
Answer (Rev. paper, Page 12 Line 373): In order to eliminate ambiguity in the article, antibiotics that were not limited in CLSI (ampicillin, doxycycline, azithromycin, clindamycin, erythromycin, linezolid, and cefazolin) were excluded from the table 3 and manuscript.
Page 11 Line 272 Table 4: According to the https://www.eucast.org/clinical_breakpoints table Clinical breakpoints - bacteria (v 12.0) - file for printing (1 Jan, 2022), there is no breakpoint for Ampicillin, please do not evaluate study results as resistant.
MIC breakpoints for Azithromycin, Clindamycin, Erythromycin, Linezolid, Cefazolin do not include in EUCAST
Answer (Rev. paper, Page 12 Line 373): As a result of the disc diffusion test, we reported the test result of Pseudomonas strains that could not be measured (value 0) as "resistant" to ampicillin Azithromycin, Clindamycin, Erythromycin, Linezolid, and Cefazolin. But, However, this point made it difficult for reviewers to understand the article. Therefore, antibiotic names that could not be included in EUCAST and CLSI were excluded from the tables and the article.
Page 13 Line 320: Please update this table according to the comments mentioned in table 4
Answer (Rev. paper, Page 12 Line 330): Corrected.
Page 13 Line 330: “bold is not necessary for Figure 5.”
Answer (Rev. paper, Page 12 Line 378): According to the review recommendation, Figure 5 was deleted.
Page 14 Line 359: The figure 5 is not necessary.
Answer (Rev. paper, Page 12 Line 378): According to the review recommendation, Figure 5 was deleted.
Page 15 Line 401: This is not a conclusion. These are result
Answer (Rev. paper, Page 13 Line 422): This sentence may report findings, but it is important to use this sentence to draw conclusions.

Reviewer 2 Report
T This manuscript describes the bacterial surface microbiota of Seabass and antibiotic susceptibility of psychrophilic Pseudomonas. Overall, the manuscript is well designed and written with an appropriate title. Some figure and table are quite small and difficult to read but can be improved. Minor recheck for spelling and format is required. I have comments/questions for the authors as follows.
The authors clearly describe the materials and methods for sample collection and molecular analysis but more details on antimicrobial susceptibility test is needed, eg. Which method has been used for MIC evaluation, and for which antibiotics?
· Some “Pseudomonas spp.” is not in italic in several point over the manuscript. Authors are suggest to recheck the format.
· Line 28 and line 64: 8 preferred antibiotics groups are mentioned but one antibiotic group in the bracket is missing.
· Line 103: full stop at the end of sentence is missing.
· Line 146: Pseudomonas Broth should not be in italic.
· Line 162-171: Authors provide 3 list of antibiotic used for Disk Diffusion test which seem to redundant to each other and could lead to confusion for reader. Authors are suggested to combine into 1 list of antibiotic or describe in more details.
· Line 204: figure 3 Seabass is misspelled.
· Figure 3 is too small and can not read text in the figure.
· Line 208: Pseudomonas is misspelled.
· Line 485: Protein is misspelled.
· Line 271: This paragraph describes the percentage of psychrophilic Pseudomonas strains (65.2%) that were sensitive to all antibiotics. However, the following paragraph also describes the same percentage as multidrug resistant psychrophilic Pseudomonas strains which is impossible. Authors are suggest to revise and correct the data.
· Results of Antibiotic susceptibility of Pseudomonas spp. by disc diffusion is confusing. In case of doripenem, 28.3% was reported in the text as resistant according to CLSI, while the same percentage also true for EUCAST. Why authors mention only results of CLSI standard in the text? I would suggest authors to select one cut off from CLSI or EUCAST for each antibiotic and clearly describe in materials and methods section in order to avoid confusion. In this case, authors will be able to report single multidrug resistant pattern for each multidrug resistant isolates. Otherwise, authors have to discuss cause of differences and compare results between CLSI and EUCAST.
· Table 4: What is the meaning of “n” in table 4?
· Results in table 4 and line 269 authors report 100% resistance to ampicillin but line 309 authors mention “we found that the Pseudomonas strains were susceptible to penicillins (AMP, TZP),…”. Authors should recheck the correctness of antimicrobial resistant results.
· Table 5 is too small.
· Line 366-367: data in the text is not comply with data in table 6. Authors are suggested to revise the data.
Author Response
Dear Reviewer 2,
We carefully considered all of the reviewer’s comments, particularly the request from one reviewer for additional data (See below). We have included several clarifications in the revised document. We would like the revised manuscript to be considered for publication in the “Foods”.
Sincerely,
Dr. Ali AYDIN
Corresponding Author
Page 1 Line 28 and line 64: 8 preferred antibiotics groups are mentioned but one antibiotic group in the bracket is missing.
Answer (Rev. paper, Page 0 Line 29): Corrected.
Page 3 Line 106: full stop at the end of sentence is missing.
Answer (Rev. paper, Page 2 Line 102): Corrected.
Page 4 Line 146: Pseudomonas Broth should not be in italic.
Answer (Rev. paper, Page 3 Line 145 and 147): Corrected
Page 7 Line 162-171: Authors provide 3 list of antibiotic used for Disk Diffusion test which seem to redundant to each other and could lead to confusion for reader. Authors are suggested to combine into 1 list of antibiotic or describe in more details.
Answer (Rev. paper, Page 10 Line 286 (Table 3), Page 12 Line 369 (Table 4) and Page 13 Line 406 (Table 5)): We organized these sections and tables.
Page 8 Line 204: figure 3 Seabass is misspelled.
Answer (Rev. paper, Page 5 Line 213): Corrected (Figure 2)
(Page 8 Line 217) Figure 3 is too small and can not read text in the figure.
Answer (Rev. paper, Page 5 and Page 6 Line 209-214): We rearranged Figures 2 and 3 by converting X-axis Tables at 90°C angle.
Page 7 Line 208: Pseudomonas is misspelled.
Answer (Rev. paper, Page 10 Line 304): Corrected
Page 18 Line 485 (Page 18 Line 521): Protein is misspelled.
Answer (Rev. paper, Page 15 Line 525): Corrected
(Page 11 Line 296) Line 271: This paragraph describes the percentage of psychrophilic Pseudomonas strains (65.2%) that were sensitive to all antibiotics. However, the following paragraph also describes the same percentage as multidrug resistant psychrophilic Pseudomonas strains which is impossible. Authors are suggest to revise and correct the data.
Answer (Rev. paper, Page 10 Line 294-296): Corrected.
(Page 11 line 292) Results of Antibiotic susceptibility of Pseudomonas spp. by disc diffusion is confusing. In case of doripenem, 28.3% was reported in the text as resistant according to CLSI, while the same percentage also true for EUCAST. Why authors mention only results of CLSI standard in the text? I would suggest authors to select one cut off from CLSI or EUCAST for each antibiotic and clearly describe in materials and methods section in order to avoid confusion. In this case, authors will be able to report single multidrug resistant pattern for each multidrug resistant isolates. Otherwise, authors have to discuss cause of differences and compare results between CLSI and EUCAST.
Answer (Rev. paper, Page 10 Line 309-315): In order to eliminate ambiguity in the article, antibiotics that were not limited in CLSI (ampicillin, doxycycline, azithromycin, clindamycin, erythromycin, linezolid, and cefazolin) and EUCAST [ampicillin, gentamycin, tobramycin (only for systemic infections isolated strains), norfloxacin, tetracycline, doxycycline, azithromycin, clindamycin, erythromycin, linezolid, and cefazolin)] were excluded from the tables and article. The article has been revised again in this respect.
(Page 12 Line 314) Table 4: What is the meaning of “n” in table 4?
Answer (Rev. paper, Page 10 Line 2927): “n” means no any limitation in CLSI and/or EUCAST standards in Table 3. We deleted all antibiotics without any limitation value in CLSI standards. And we added a footnote about “n”.
Results in table 4 and line 269 authors report 100% resistance to ampicillin, but line 309 authors mention “we found that the Pseudomonas strains were susceptible to penicillins (AMP, TZP),…”. Authors should recheck the correctness of antimicrobial resistant results.
Answer (Rev. paper, Page 10 Line 289): Table 4 changed to Table 3. In order to eliminate ambiguity in the article, antibiotics that were not limited in CLSI (ampicillin, doxycycline, azithromycin, clindamycin, erythromycin, linezolid, and cefazolin) and EUCAST [ampicillin, gentamycin, tobramycin (only for systemic infections isolated strains), norfloxacin, tetracycline, doxycycline, azithromycin, clindamycin, erythromycin, linezolid, and cefazolin)] were excluded from the tables and article. The article has been revised again in this respect.
(Page 13 Line 350) Table 5 is too small.
Answer (Rev. paper, Page 12 Line 373): Table 5 changed to Table 4. Corrected.
(Page 15 Line 420) Line 366-367: data in the text is not comply with data in table 6. Authors are suggested to revise the data.
Answer (Rev. paper, Page 13 Line 407): Table 6 changed to Table 5. Corrected

Reviewer 3 Report
The manuscript is generally poorly written and needs an extensive reconstruction and language editing. There are many examples of wrong wording.
The results could have discussed better, it is for example not mentioned that the bacteria they have found in the samples are very common environmental bacteria that are expected to be found on the fish.
Did you test if the ice that the fish was transported in contained bacteria? How do you know if the bacteria came from the ice or from the fish?
You should be discuss that Pseudomonas spp. are naturally very tolerant against some of the antibiotics you have tested. That should be mentioned. Can you also explain why you are testing drugs e.g. clindamycin that is designed for G+ bacteria and are known to not work for Pseudomonas spp.?
You found enterococci in the fish samples. What could that mean for food safety?
It would also be valuable for readers who are not very familiar with fish farming in Turkey to know which of the antibiotics tested are used in the fish farming in your country and how commonly they are used (e.g., tonnes annually).
You call the Pseudomonas isolates psycrophilic -I think its more appropriate to call them psycrothrophic.
If all Pseudomonas strains you analysed are psycrothrophic you could write that in the beginning so that you dont need to mention it over and over again.
You can find other comments and suggestions in the attached manuscript.

Author Response
Dear Reviewer 3,
We carefully considered all of the reviewer’s comments, particularly the request from one reviewer for additional data (See below). We have included several clarifications in the revised document. We would like the revised manuscript to be considered for publication in the “Foods”.
Sincerely,
Dr. Ali AYDIN
Corresponding Author
Page 1 Line 15: This sentence doesn not make sense.
Answer (Rev. paper, Page 0 Line 16): Corrected
Page 1 Line 16: Dominant species in what?
Answer (Rev. paper, Page 0 Line 17): This sentence was deleted.
Page 1 Line 17: This sentence should come after "In this study"
Answer (Rev. paper, Page 0 Line 18): Corrected
Page 1 Line 20: must define abbreviation the first time its used.
Answer (Rev. paper, Page 0 Line 25-26): According to the Reviewer recommendation, we used the full name of the abbreviations at the first mention in the Abstract and text ( Page 1 Line 73, and 74).
Page 1 Line 23: Do you mean that 46 out of 96 seabass samples contained Pseudomonas spp.?
Answer (Rev. paper, Page 9 Line 270): Yes. While metagenomics method can highlight information on the bacterial community, it cannot differentiate between live and dead bacteria. Thus, to isolate live Pseudomonas strains (n=46) represented by 48%, we complemented metagenomics method with NGS technique. Futhermore, this can allow the detection of non-viable and viable DNA fragments [39]. (39: Chiu, C.Y.; Miller, S.A. Clinical metagenomics. Microb. Genom.2019, 20, 341-355.
Page 1 Line 20: Abbreviations should be explained the first time they are used in a text
Answer (Rev. paper, Page 1 Line 25-26): According to the reviewer recommendation, we used the full name they are used in the manuscript (for example Page 1 Line 73 and 74).
Page 1 Line 25: Rephrase: For example: Pseudomonas strains were tested for susceptibility to a total of 18 antibiotic from eight different groups of antibiotics.
Answer (Rev. paper, Page 1 Line 27): Changed as “Pseudomonas strains were tested for susceptibility to 11 antibiotics (piperacillin-tazobactam, gentamicin, tobramycin, amikacin, doripenem, meropenem, imipenem, levofloxacin, ciprofloxacin, norfloxacin, and tetracycline) from 5 different groups of antibiotics (penicillin, aminoglycosides, carbapenems, fluoroquinolones, and tetracyclines)”.
Page 1 Line 30: 28.3% cannot be most of the strains, its only close to 1/3
Answer (Rev. paper, Page 0 Line 32): This sentence changed.
Page 1 Line 31: delete “total”
Answer (Rev. paper, Page 1 Line 32): This sentence changed.
Page 1 Line 31: This is not surprising as its already well known that most aerobic Gram-negative bacteria are resistant to clindamycin including Pseudomonas spp.
The same, here -its well known that pseudomonas spp. are resistant to ampicillin.
You should discuss in the article that Pseudomonas is naturally resistant against clindamycin and ampicillin and that the 100% reistance is thus expected.
Answer (Rev. paper, Page 0 Line 33): According to the second reviewer’s recommendation, antibiotics without limits such as clindamycin and ampicillin etc. in EUCAST and CLSI standards were removed from the manuscript to avoid confusion. On the other hand, reviewer is right in this comment. However, since those sections have been removed from the article, the answering situation has disappeared.
Page 1 Line 32: delete “first”
Answer (Rev. paper, Page 0 Line 33): Deleted
Page 1 Line 32: In what sense is it dynamic?
Answer (Rev. paper, Page 0 Line 34): This word is deleted
Page 1 Line 33: CLSI and EUCAST are methods used for therapy testing. It should be discussed in the article why you used it for testing fish isolates.
Answer (Rev. paper, Page 1 Line 73-77): We added related sentence in revised paper. (Commonly, EUCAST and CLSI standards are frequently used in this article to detect antibiotic resistance in foodborne bacteria, including fish. In addition, antibiotics are frequently used in the treatment of diseases in fish farming. Long-term and high-dose use of antibiotics also creates a resistance problem in bacteria present in foods).
Page 1 Line 33: "involved" is the wrong word here. Delete it.
Answer (Rev. paper, Page 1 Line 34): Deleted
Page 1 Line 34: I would rather call the bacteria you are working with psycrophilic instead of psycrothrophic
Answer (Rev. paper, Page 1 Line 35): We agreed. We used also “psychrophilic” in the manuscript.
Page 2 Line 49: “most popular in comparison to what
Answer (Rev. paper, Page 1 Line 48): We deleted this sentence according to the first reviewer recommendation.
Page 2 Line 54: “One billion 376 million dollars” ????
Answer(Rev. paper, Page 1 Line 50): We deleted this sentence according to the first reviewer recommendation. But, the data is true (https://www.tarimturk.com.tr/haber-su-urunleri-ihracatinda-cipura-levrek-ve-turk-somonu-basi-cekti-5467) (In Turkish).
Page 2 Line 57: provide important knowledge of the composition of the microbiota?
Answer (Rev. paper, Page 1 Line 52): Related to this article, provides us with reliable data on “skin microbiota of fish”.
Page 2 Line 57 and 59: Use “can”
Answer(Rev. paper, Page 1 Line 57): We changed these sentences.
Page 2 Line 62: There already studies that investigate the microbiota of seabass- you should be more specific here on exactly what knowledge is missing?
Answer(Rev. paper, Page 1 Line 70-87): In this article, for the first time, it belongs to Turkey, which is the country that produces the most sea bass in the world; There is information about the fish skin microbiota originating from the Aegean Sea and antibiotic resistance in the dominant Pseudomonas strains here. In addition, instead of choosing EUCAST or CLSI for detecting antibiotic resistance, revealing data on antibiotic resistance related to Pseudomonas in both standards increases the specificity of this study.
Page 2 Line 64: The bacteria are not pathogenic for the fishery industry but knowledge on the presence of pathogenic bacteria in the food raw material could help the industry to target preventative measures.
Answer (Rev. paper, Page 1 Line 57-59): We agree with this comment. We changed necessary points, according to the comment in manuscript
Page 2 Line 68: delete “primary pathogen”
Answer (Rev. paper, Page 1 Line 61): Deleted
Page 2 Line 69: Do you mean benefical for human health???? “causes human health”
Answer (Rev. paper, Page 2 Line 63): Deleted
Page 2 Line 71: The products do not get sick!
Answer (Rev. paper, Page 1 Line 64-66): This sentence was rewritten.
Page 2 Line 72: Consequently
Answer (Rev. paper, Page 1 Line 67): Corrected
Page 2 Line 85: “preffered in what sense?
Answer (Rev. paper, Page 1 Line 73): Deleted
Page 2 Line 93: Use “Aquaculture companies”
Answer (Rev. paper, Page 2 Line 104): We used “aquaculture” instead of “fish” all the paper.
Page 2 Line 95: Was the ice tested for the presence of bacteria. IS there a risk that the ice has contaminated the fish.
Answer (Rev. paper, Page 1 Line 95): Aseptic ice was used to preserve the fish in this study. This statement has been added to the article.
Page 3 Line 97: In the fish or in the box?
Answer (Rev. paper, Page 2 Line 94): “In the fish”
Page 3 Line 101: Specify netter which media
Answer (Rev. paper, Page 2 Line 99): This media mean “transport liquid medium”. We added in text.
Page 3 Line 103: Delete “analyzed”
Answer (Rev. paper, Page 2 Line 102): We used “analyses” instead of “analyzed”.
Page 4 Line 146: each for what?
Answer (Rev. paper, Page 3 Line 145): We used “liquid medium” instead of “broth”.
Page 7 Line 186: Use “in the” instead of “among”
Answer (Rev. paper, Page 3 Line 174): Changed
Page 7 Line 191: Use “was” instead of “were”
Answer (Rev. paper, Page 4 Line 180): Changed
Page 7 Line 187: Delete “the other identified bacteria with high read numbers at genus level
Answer (Rev. paper, Page 4 Line 194): We didn’t delete these sentences. Because these data are important for the paper related to Figure 3.
Page 7 Line 195: Suggestion to rephrasing: ....was the dominant genera in MAP-stored seabass fillets and its relative abundance declined dramatically towards the end of the...
Answer (Rev. paper, Page 5 Line 202): Corrected
Page 7 Line 200 and 201: Use “and is thus” instead of “being in parallel”
Answer (Rev. paper, Page 5 Line 207): Corrected
Page 7 Line 198: I don’t think Acinetobacter found anything.
Answer (Rev. paper, Page 5 Line 206): We used this sentence from reference Syropoulou et al. (2022). Posssible can be checked the reference (Ref No 27: Syropoulou, F.; Anagnostopoulos, D.A.; Parlapani, F.F.; Karamani, E.; Stamatiou, A.; Tzokas, K.; Nychas, G.-J.E.; Boziaris, I.S. Microbiota Succession of Whole and Filleted European Sea Bass (Dicentrarchus labrax) during Storage under Aerobic and MAP Conditions via 16S rRNA Gene HighThroughput Sequencing Approach. Microorganisms 2022, 10, 1870).
Page 7 Line 200: Use “important” instead of “strong”
Answer (Rev. paper, Page 5 Line 206): Corrected
Page 7 Line 200: Delete “However”
Answer (Rev. paper, Page 5 Line 206): Deleted
Page 7 Line 204: Figure 3: Could be better to show this image in landscape view.
Answer (Rev. paper, Page 5 Line 209): If it is possible, we want to use this format in the manuscript.
Page 8 Line 205: In this materials and methods part, already mention that you used this method to characterize the isolates down to the species level. Therefore you don’t need repeat this so many times in the text.
Answer (Rev. paper, Page 6 Line 217): Repeated sentence are deleted
Page 8 Line 207: Delete “In this context”
Answer (Rev. paper, Page 8 Line 219): Deleted
Page 8 Line 209: What do the presence of enterococcus in the seabass samples indicate and why could this be associated with a food safety risk?
Answer (Rev. paper, Page 7 Line 222): Enterococci are now increasingly featured in publications as food pathogens. Antibiotic resistant enterococci strains, especially vancomycin resistance, are important for public health. Therefore, we presented Enterococcus results in this article.
Page 8 Line 216: Delete “most diverse species”
Answer (Rev. paper, Page 7 Line 225): Deleted
Page 8 Line 216: lowest species diversity
Answer (Rev. paper, Page 7 Line 227): Corrected
Page 8 Line 216: wrong wording?
Answer (Rev. paper, Page 7 Line 227): Corrected
Page 8 Line 221: Use “was”
Answer (Rev. paper, Page 7 Line 231): Corrected
Page 8 Line 221: Delete “at the species level”
Answer (Rev. paper, Page 7 Line 231): Deleted
Page 8 Line 223: It has already been mentioned in the material and methods part you characterized isolates down to species level by using the metabarcoding technique. Therefore, you don’t need to repeat it here.
Answer (Rev. paper, Page 7 Line 233): Deleted
Page 8 Line 224: I suggest you change the wording here to “bacteria belonging to the family Vibrionaceae.
Answer (Rev. paper, Page 7 Line 239): Changed
Page 10 Line 232 and 234: Use “to be” instead of “at“
Answer (Rev. paper, Page 9 Line 253): Changed
Page 10 Line 244: Dominant player. Changed to predominant genera
Answer (Rev. paper, Page 10 Line 256): Changed
Page 10 Line 245: This sentence does not make sense. Consider changing it to … “upon storage at cold temperature, pscycothrophic species proliferated slowly and were predominat over sesophilic species.
Answer (Rev. paper, Page 9 Line 258-259): Corrected.
Page 10 Line 247: Use “such as” instead of “of”
Answer (Rev. paper, Page 9 Line 260): Changed
Page 10 Line 247: Add “spp.”
Answer (Rev. paper, Page 9 Line 260): Corrected.
Page 10 Line 247: Use “dominated” instead of “which comprised”??
Answer (Rev. paper, Page 9 Line 258): Corrected.
Page 10 Line 248: Pseudomonas spp. were found from the beginning of shelf life wile both Pseudomonas and Shewanella spp. were found at the end of self life.
Answer (Rev. paper, Page 9 Line 260): We agreed reviewer comments. However, here we used the conclusion and interpretation of Syropoulu et al (38). The expression belongs to them.
Page 10 Line 249: Table 2; In English, comma is just used to separate numbers higher that 999. Use (.) instead (,).
Answer (Rev. paper, Page 9 Line 273): Corrected.
Page 10 Line 257: As the reason for isolating 48% (n=46) live Pseudomonas strains within the scope of 256 the study; The detection of non-viable DNA fragments as well as DNA structures that do 257 not belong to living bacteria with NGS method and metabarcoding analysis is shown [35]
Answer (Rev. paper, Page 9 Line 269-272): While metagenomics method can highlight information on the bacterial community, it cannot differentiate between live and dead bacteria. Thus, to isolate live Pseudomonas strains (n=46) represented by 48%, we complemented metagenomics method with NGS technique. Furthermore, this can allow the detection of non-viable and viable DNA fragments [39].
Page 11 Line 267: Use “of the”
Answer (Rev. paper, Page 9 Line 283): Corrected.
Page 11 Line 267: Add “isolates”
Answer (Rev. paper, Page 9 Line 283): Corrected.
Page 11 Line 274: Use “A total of” instead of “In total”
Answer (Rev. paper, Page 10 Line 293): This sentence was rewritten.
Page 11 Line 277 and 278 : Delete “detected as” and “determined to be”
Answer (Rev. paper, Page 10 Line 293 and 294): This sentence was rewritten.
Page 11 Line 280: Rephrase. Suggestion: Additionally, all six psycrophilic Pseudomonas strains, from different fish farms in Izmir, were resistant to...
Answer (Rev. paper, Page 10 Line 304): Corrected.
Page 11 Line 286: Move (n= 46)
Answer (Rev. paper, Page 10 Line 310): This sentence was rewritten.
Page 12 Line 306: Use “particularly” instead of “respectively”
Answer (Rev. paper, Page 11 Line 328: Corrected.
Page 12 Line 307: Use “to threat infection in animals and humans”
Answer (Rev. paper, Page 11 Line 329): Corrected.
Page 13 Line 329: The Petri dishes are not resistant -the bacteria that grows on them can be
Answer (Rev. paper, Page 11 Line 341): According to reviewer 2 comments, we deleted Figure 5 and this sentence.
Page 13 Line 334: Rephrasing needed. Suggestion: Similarly, psycrotrophic Pseudomonas strains resistant to most antibiotics (n=9) were isolated from 5 different fish farms (…), X.
Answer (Rev. paper, Page 11 Line 346-348): Corrected.
Page 13 Line 339: “Pseudomonas” should be italicized.
Answer: (Rev. paper, Page 11 Line 351): Corrected.
Page 13 Line 347: ???
Answer (Rev. paper, Page 11 Line 353): Corrected. …(different fish farms)
Page 13 Line 350: Which fish farm ???
Answer (Rev. paper, Page 11 Line 354): We deleted this sentence according to the reviewer 2. Because no limitation in CLSI for these antibiotics.
Page 13 Line 354: Use “often” instead of “is still in many cases”
Answer (Rev. paper, Page 11 Line 356): Corrected.
Page 13 Line 356: Use “challenging” instead of “facing”
Answer (Rev. paper, Page 11 Line 358): Corrected.
Page 13 Line 356:”This is not true”
Answer (Rev. paper, Page 11 Line 358): Corrected. As “some studies” according to reference 44 (Rigos et al., 2021).
Page 14 Line 366:”This sentence does not make sense.
Answer (Rev. paper, Page 13 Line 415-417): Deleted (We reorganized this section).
Page 14 Line 368: Delete “resistant”
Answer (Rev. paper, Page 13 Line 419): Deleted (We reorganized this section).
Page 14 Line 370: Use “of them”
Answer (Rev. paper, Page 13 Line 419): Deleted (We reorganized this section).
Page 14 Line 373: Add “resistance”
Answer (Rev. paper, Page 13 Line 422): Deleted (We reorganized this section).
Page 14 Line 375: Delete “Besides”
Answer (Rev. paper, Page 13 Line 424): Deleted (We reorganized this section).
Page 15 Line 391: Strange wording
Answer (Rev. paper, Page 14 Line 449): We are rewritten section 3.5.
Page 15 Line 391: Use “fish” instead of “bacterial”
Answer (Rev. paper, Page 13 Line 404): Corrected. “We add the name of the fish “Nile Tilapia”.
Page 15 Line 409: Has this ever been shown to occur?
Answer (Rev. paper, Page 15 Line 434-436): This sentence rewritte

Reviewer 4 Report
In their paper, the authors describe the bacterial microbiota on the skin of seabass with a focus on determining the resistance of bacteria of the genus Pseudomonas. This is an interesting and very timely topic, especially given the pervasive concern about antibiotic resistance in the environment, which can also affect human health.
1. From that what the authors have written, it appears that this is a comprehensive study in which the authors have taken swabs from the mouth and gills in addition to the skin, but for this manuscript they have chosen to present only the microbiota from the skin of seabass. In this regard, it is recommended to delete from the manuscript that swabs were taken from mouth and gills, as the results for this are not presented or discussed in this paper (in Sampling).
2. In the same paragraph "in Sampling" there is a space and two commas after C2, which should also be corrected.
3. ‘’Biochemical tests such as Gram staining, oxidase test, catalase test and fluorescence properties were applied to confirm Pseudomonas strains [9,18].’’ How were the fluorescence properties determined?
4. Muller Hinton agar was manufactured by Oxoid, and whose discs were they?
5. Why did the authors choose 44+4h incubation rather than 24h when determining resistance? According to which protocol?
6. "P. lundensis was identified in all seabass samples.'' I ask the authors to present this better. it is not visible from the results and Tables.
7. "Elbehiry et al [27] reported that in meat samples P. lundensis was the predominant species.’’ According to what the authors have written, it looks like it is fish meat, but if it is not fish meat, then it should be written clearly.
8. ‘’Berggren et al [6] have recently demonstrated that the community composition of the fish skin microbiomes a) was different overall from the microbial community in the surrounding water, b) did not differ between dorsal and ventral body sites within hosts, c) varied considerably among hosts, and d) differed according to the host population.’’ According to that, did the authors determine the microbial community in the surrounding water?
9. Vibrio ordalii, same situation as P. lundensis, where is V. ordalii in the results and tables?
10. What is the reason that Vibrio is not the most dominant genus in this study and has been outcompeted by other genera, according to the authors?
11. I please the authors to better label the results from Table 2, as they did not follow the information in the text. "In total 46 seabass swab samples (48%) were positive for psychrophilic Pseudo-monas strains isolated with the conventional ISO method [18] [Table 2]." If 48 is the percentage in Table 2, then it should be marked in the table how it is presented "number of live Pseudomonas isolates based on NGS positives" and not %?
12. Is it possible for the authors to separate the resistance results of P. lundensis from those of psychrophilic Pseudomonas?
Author Response
Dear Reviewer 4,
We carefully considered all of the reviewer’s comments, particularly the request from one reviewer for additional data (See below). We have included several clarifications in the revised document. We would like the revised manuscript to be considered for publication in the “Foods”.
Sincerely,
Dr. Ali AYDIN
Corresponding Author
- From that what the authors have written, it appears that this is a comprehensive study in which the authors have taken swabs from the mouth and gills in addition to the skin, but for this manuscript they have chosen to present only the microbiota from the skin of seabass. In this regard, it is recommended to delete from the manuscript that swabs were taken from mouth and gills, as the results for this are not presented or discussed in this paper (in Sampling).
Answer (Page 2, Line 99): According to the reviewer suggestion, “mouth and gills” were deleted from the paper.
- In the same paragraph "in Sampling" there is a space and two commas after C2, which should also be corrected.
Answer (Page 1, Line 93): Corrected and two commas deleted.
- ‘’Biochemical tests such as Gram staining, oxidase test, catalase test and fluorescence properties were applied to confirm Pseudomonas strains [9,18].’’ How were the fluorescence properties determined?
Answer (Page 3, Line 152): Corrected. According to the Oxoid, manufacture media catalog under Pseudomonas agar base Technique (http://www.oxoid.com/UK/blue/prod_detail/prod_detail.asp?pr=CM0559&c=UK&lang=EN); It is stated that after the CFC supplement (SR0103) added to the medium, the samples can be detected under UV light (365 nm) after 24-48 hours at 25° C incubation. Accordingly, after 48 hours of incubation at 22-24 C, we detected the fluorescence feature by placing the petri plates under a UV light.
- Muller Hinton agar was manufactured by Oxoid, and whose discs were they?
Answer (Page 3, Lines 157-162): The names of the companies producing the discs and the product codes have been added to the article.
- Why did the authors choose 44+4h incubation rather than 24h when determining resistance? According to which protocol?
Answer (Page 3, Line 156): Corrected. In the first version of the article, the incubation period was written as 24h, while in the revision version it was written as 44+4h inadvertently.
- "P. lundensiswas identified in all seabass samples.'' I ask the authors to present this better. it is not visible from the results and Tables.
Answer (Page 7, Line 217): According to the NGS analysis results are examined; The % values of P. lundensis presence at the species level in sea bass skin samples are presented, and it is included in the data that P. lundensis is found in all of the examined samples. In addition, the presence of psychrophilic Pseudomonas was examined with classical methods within the scope of the project, and no identification at the species level (eg P. lundensis, P. aeruginosa etc.) was made. Accordingly, NGS analysis result data is presented (Data can be submitted if the reviewer requests it).
- "Elbehiry et al [27] reported that in meat samples P. lundensiswas the predominant species.’’ According to what the authors have written, it looks like it is fish meat, but if it is not fish meat, then it should be written clearly.
Answer (Page 7, Line 218): Elbehiry et al [27] reported that in red meat samples P. lundensis was the predominant species. We have not yet come across a publication stating that P. lundensis is dominant in fish meats.
- ‘’Berggren et al [6] have recently demonstrated that the community composition of the fish skin microbiomes a) was different overall from the microbial community in the surrounding water, b) did not differ between dorsal and ventral body sites within hosts, c) varied considerably among hosts, and d) differed according to the host population.’’ According to that, did the authors determine the microbial community in the surrounding water?
Answer (Page 7, Line 230): We didn't search microbial community in the surrounding water. However, we consider it important to present the research results of Bergran et al. (6).
- Vibrioordalii, same situation as P. lundensis, where is V. ordalii in the results and tables?
Answer (Page 7, Line 239): When NGS analysis results are examined; In only 3 samples of sea bass skin, foodborne Vibrio spp. existence has been demonstrated. The % values of Vibrio ordalii presence at the species level in sea bass skin samples are presented in NGS result data. In addition, the presence of psychrophilic Pseudomonas was examined with classical methods within the scope of the project, and no identification at the species level of Vibrio spp.
- What is the reason that Vibrio is not the most dominant genus in this study and has been outcompeted by other genera, according to the authors?
Answer (Page 7, Line 234-240): Accordint to our knowledge, the quality of the sea water in the region where the sea bass grow can be shown as the reason why Vibrio spp is not a dominant genus.
- I please the authors to better label the results from Table 2, as they did not follow the information in the text. "In total 46 seabass swab samples (48%) were positive for psychrophilic Pseudo-monas strains isolated with the conventional ISO method [18] [Table 2]." If 48 is the percentage in Table 2, then it should be marked in the table how it is presented "number of live Pseudomonasisolates based on NGS positives" and not %?
Answer (Page 9 Line 273): Corrected :
- Is it possible for the authors to separate the resistance results of P. lundensisfrom those of psychrophilic Pseudomonas?
Answer (Page 10, Line 286): Within the scope of NGS results and antibiotic susceptibility tests, it is not possible to separate the resistance results of P. lundensis from those of psychrophilic Pseudomonas. In addition, the presence of psychrophilic Pseudomonas was examined with classical methods within the scope of the project, and no identification at the species level (eg P. lundensis, P. aeruginosa etc.) was made.

Round 2
Reviewer 1 Report
Dear Authors,
You have made a good edition of the paper and I have no doubts of the results.
Author Response
Thank you for reviewing our article.
Reviewer 3 Report
please see the attachment

Author Response
Dear Reviewer 3,
We carefully considered all of your comments, particularly the request from one reviewer for additional data (See below). We have included several clarifications in the revised document. We would like the revised manuscript to be considered for publication in the “Foods”.
Sincerely,
Dr. Ali AYDIN
Corresponding Author
P.S: Page numbering in the manuscript has been adjusted to make the manuscript start with page 1.
Page 1 Line 23: Do you mean that you were able to culture pseudomonas from 46 of the samples that were positive for these genera by NGS?
Answer (Rev. paper, Page 1 Line 23): Yes. Yes. While the metagenomics method can highlight information on the bacterial community, it cannot differentiate between live and dead bacteria. Thus, to isolate live Pseudomonas strains (n=46) represented by 48%, we complemented the metagenomics method with the NGS technique. Furthermore, this can allow the detection of non-viable and viable DNA fragments [39]. (39: Chiu, C.Y.; Miller, S.A. Clinical metagenomics. Microb. Genom.2019, 20, 341-355.
Page 1 Line 26: Were these isolates really psycrophlic? How was this tested?
Answer (Rev. paper, Page1 Line 23): The suggestion of the referee regarding psychrophilic and psychrotrophic expressions was reviewed by the authors. According to the reviewer’s recommendation, the authors used “psychrotrophic” instead of “psychrophilic” in all the manuscript.
Page 1 Line 34: Use “of” instead of “in/on” This sentence should come after "In this study"
Answer (Rev. paper, Page 1 Line 33): Corrected.
Page 1 Line 42: Delete “in an efficient way”.
Answer (Rev. paper, Page 1 Line 42): Deleted.
Page 2 Line 48: Delete “from”.
Answer (Rev. paper, Page 2 Line 48): Deleted.
Page 2 Line 49: Suggestion to rephrasing: Turkey and Greece are the primary producers, while Spain, France, Italy, Greece, and Turkey are the primary consumers.
Answer (Rev. paper, Page 2 Lines 47-48): Corrected.
Page 2 Line 51: Consider inserting this section to row 78.
Answer (Rev. paper, Page 2 Line 70-71): Corrected. This section has been added to line 78.
Page 2 Line 70: Delete “that”.
Answer (Rev. paper, Page 2 Line 66): Deleted.
Page 2 Line 76: Consider moving this sentence to a place in the text where it fits better
Answer (Rev. paper, Page 2 Line 74): This sentence has already been mentioned in the text above (lines 62-65) indirectly. Therefore, we deleted this sentence.
Page 2 Line 81-86: You mix between writing “were performed” and “will be used” it looks much better if you are more consistent.
Answer (Rev. paper, Page 2 Line 77 and line 82): Corrected.
Page 2 Line 82: “?”
Answer (Rev. paper, Page 2 Line 78): corrected. We added “against”.
Page 3 Line 127: Use “purified”
Answer (Rev. paper, Page 3 Line 123): Corrected.
Page 4 Line 142 and line 154: Why italics”
Answer (Rev. paper, Page 4 Line 138 and 150): According to the Author’s Guidelines of Foods Journal all subtitles should be italics such as 1.1, 2.2, 3.3 except sub subtitles such as 2.21.; 2.2.2.
Page 4 Line 165: consider rephrasing. for example: ....was applied to determine the MIC of Pseudomonas strains that were found to be resistant to antibiotics in the disc diffusion test
Answer (Rev. paper, Page 4 Lines 162--164): Corrected
Page 4 Line 170: long sentence consider rephrasing
Answer (Rev. paper, Page 4 Line 167): This sentence was rewritten.
Page 4 Line 170: Use “increasing” instead of “enrichment”
Answer (Rev. paper, Page 4 Line 169): Corrected.
Page 4 Line 177: “and their relative abundance”
Answer (Rev. paper, Page 4 Line 171-172): Corrected.
Page 4 Line 179: Change as “complex bacterial community” and delete “and other pathogens”.
Answer (Rev. paper, Page 4 Line 173): Corrected.
Page 5 Line 186: I think you should elaborate on the results presented in this table in the results part
Answer (Rev. paper, Page 5 Line 180): We tried to detail the results related to table 1 between lines 184-190 and page 8 lines 222-226.
Page 5 Line 189: present thousands consistently throughout the text
Answer (Rev. paper, Page 5 Line 186): We agreed with reviewers' comments.
Page 5 Line 90: change as “the phylum Proteobacteria”
Answer(Rev. paper, Page 5 Line 186): Corrected.
Page 55 Line 191: “The skin microbiota samples also contained bacteria belonging to the phyla ...”
Answer (Rev. paper, Page 5 Line 186-187): We changed this sentence according to the reviewer’s recommendation.
Page 5 Line 192: “consider rephrasing e.g., Pseudomonas was the dominant genera among the 96...”
Answer (Rev. paper, Page 5 Line 188): We changed this sentence according to the reviewer’s recommendation.
Page 5 Line 194: “were also among the most prevalent genera”
Answer(Rev. paper, Page 5 Line 189-190): We changed this sentence according to the reviewer’s recommendation.
Page 5 Line 195: from fish skin samples?
Answer (Rev. paper, Page 5 Line 192): No. These results are presented from seawater samples. Corrected.
Page 5 Line 196: strange wording
Answer (Rev. paper, Page 5 Line 191-192): Corrected.
Page 5 Line 197: If you did the last study it was you that corroborated the results from the previous study
Answer (Rev. paper, Page 5 Line 193-194): According to the reviewer’s recommendation, “corroborated” changed as “similar to”.
Page 5 Line 198: your study or the other study you refer to in the previous sentence?
Answer (Rev. paper, Page 5 Line 194-197): We have rewritten this sentence according to the reviewer’s recommendation.
Page 6 Line 201: change to "an important spoilage bacterium of seafood"
Answer (Rev. paper, Page 6 Line 198-199): We have rewritten this sentence according to the reviewer’s recommendation.
Page 6 Line 204: use “frequently”
Answer (Rev. paper, Page 7 Line 200): Corrected.
Page 7 Line 214: Also difficult to read the text in this figure.
Answer (Rev. paper, Page 7 Line 208): Figure 3 is difficult to read because it is included in the journal template format. However, we will return the shape in “tif” format to the online system during the printing phase.
Page 8 Line 217: did you identify P. lundensis based on the NGS data or by other analyses?
Answer (Rev. paper, Page8 Line 213): According to the NGS analysis results are examined; The % values of P. lundensis presence at the species level in sea bass skin samples are presented, and it is included in the data that P. lundensis is found in all of the examined samples. In addition, the presence of Pseudomonas was examined with classical methods within the scope of the project. No identification at the species level (eg P. lundensis, P. aeruginosa etc.) was made. Accordingly, NGS analysis result data is presented.
Page 8 Line 220-223: Could this be an indication of fecal contaminaton of the water the fish live in
Answer (Rev. paper, Page 8 Line 220-221): We added related sentence according to the reviewer’s recommendation.
Page 8 Line 227: Could you indicate what genera you found in the clusters? You show the figure but I wish you could elaborate more on the results it shows
Answer (Rev. paper, Page 8 Line 226): This situation is partially detailed in Figure 4.
Page 8 Line 228: Do you mean that the composition of the microbiota did not cluster according to sampling location?
Answer (Rev. paper, Page 8 Line 226-229): We agreed reviewers comments.
Page 8 Line 235: did you identify the bacterium down to the species level based on sequence data or other tests?
Answer (Rev. paper, Page 8 Line 233): According to the NGS analysis results are examined; V. ordalii presence at the species level in sea bass skin samples is presented.
Page 8 Line 238: Do you mean infections?
Answer (Rev. paper, Page 8 Line 234): Corrected. We use infections” instead of “problems”.
Page 10 Line 266: Are you sure you mean psychrophilic and not psychrotrophic?
Answer (Rev. paper, Page 10 Line 262): According to the reviewer’s recommendation, we used “psychrotrophic” instead of “psychrophilic” all the paper.
Page 10 Line 269: do you mean cultivation-based methods?
Answer (Rev. paper, Page 10 Line 265): According to the reviewer’s recommendation, this word changed to “cultivation-based”.
Page 10 Line 270: what do you mean with non-viable DNA fragments?
Answer (Rev. paper, Page 10 Line 266): According to the reviewer’s recommendation, “non-viable” deleted.
Page 10 Line 274: It is not clear to me what you mean here. Do you mean how many samples you were able to cultivate Pseudomonas from out of those that were positive for this genus in the NGS analysis?
Answer (Rev. paper, Page 10 Line 273): We agreed reviewer’s comment. Therefore we removed all the percentages in Table 2 and we rewritten some subtitles in Table 2.
Page 10 Line 281: The text about the antibiotic resistance patterns feels overwhelming. Is there a way you could make the text more concise?
Answer (Rev. paper, Page 10 Line 277): We can try to reduce “antibiotic resistance” words in the text.
Page 11 Line 293: If all your pseudomonas isolates were psychrophilic you can mention it once so that you dont need to repeat it over and over again.
Answer (Rev. paper, Page 11 Line 288): According to the reviewer’s recommendation, “psychrophilic” were deleted.
Page 11 Line 296: Susceptible to what? All antibiotics or just some?.
Answer (Rev. paper, Page 11 Line 291): According to the reviewer’s recommendation, added “to all antibiotics”.
Page 12 Line 319-321: Consider rephrasing e.g., Pseudomonas spp. have been identified as primarily invasive or opportunistic pathogens for many organisms, and this genus has also grown in importance in terms of antimicrobial resistance.
Answer (Rev. paper, Page 12 Line 311-313): This sentence was rewritten per the reviewer’s recommendation.
Page 12 Line 342-345: Delete you don’t need to add references to these methods more than when you first mention them.
Answer (Rev. paper, Page 12 Line 334-337): According to the reviewer’s recommendation, the reference number was deleted.
Page 13 Line 381: Delete “and”
Answer (Rev. paper, Page 13 Line 376): Deleted
Page 13 Line 384: what about these? Do you refer to these isolates in the sentence before?
Answer (Rev. paper, Page 13 Line 381): We have also referred to these isolates in the previous sentences of the article.
Page 13 Line 395: where were these strains from? Foods, patients...?
Answer (Rev. paper, Page 13 Line 390): We find in this surveillance report about P. aeruginosa origin from human patients (We added in this paper). EARS-Net-report-2017-update-jan-2019.pdf.

Reviewer 4 Report
The authors accepted the suggested corrections.
Author Response
Thank you for reviewing our article.